# Lineage Replacement and Genetic Changes of Four HR-HPV Types during the Period of Vaccine Coverage: A Six-Year Retrospective Study in Eastern China

**DOI:** 10.3390/vaccines12040411

**Published:** 2024-04-12

**Authors:** Wenjie Qu, Chen Hua, Yaping Wang, Yan Wang, Lu Zhang, Zhiheng Wang, Wenqian Shi, Fang Chen, Zhiyong Wu, Qian Wang, Lu Lu, Shibo Jiang, Long Sui, Yanyun Li

**Affiliations:** 1Department of Gynecology and Obstetrics, Obstetrics and Gynecology Hospital of Fudan University, Shanghai 200011, China; ququanquanns@163.com (W.Q.);; 2Shanghai Institute of Infection Disease and Biosecurity, Shanghai Medical College, Fudan University, Shanghai 200032, China; hua_chen@fudan.edu.cn (C.H.);; 3Qingpu Branch of the Yangtze River Delta Integrated Demonstration Zone, Obstetrics and Gynecology Hospital of Fudan University, Shanghai 200090, China; 4Shanghai Public Health Clinical Center, Key Laboratory of Medical Molecular Virology (MOE/NHC/CAMS), Shanghai Institute of Infectious Disease and Biosecurity, School of Basic Medical Sciences, Fudan University, Shanghai 201508, China

**Keywords:** human papillomavirus, vaccine, lineage replacement, vaccine escape

## Abstract

Objective: This study aimed to provide clinical evidence for lineage replacement and genetic changes of High-Risk Human Papillomavirus (HR-HPV) during the period of vaccine coverage and characterize those changes in eastern China. Methods: This study consisted of two stages. A total of 90,583 patients visiting the Obstetrics and Gynecology Hospital of Fudan University from March 2018 to March 2022 were included in the HPV typing analysis. Another 1076 patients who tested positive for HPV31, 33, 52, or 58 from November 2020 to August 2023 were further included for HPV sequencing. Vaccination records, especially vaccine types and the third dose administration time, medical history, and cervical cytology samples were collected. Viral DNA sequencing was then conducted, followed by phylogenetic analysis and sequence alignment. Results: The overall proportion of HPV31 and 58 infections increased by 1.23% and 0.51%, respectively, while infection by HPV33 and 52 decreased by 0.42% and 1.43%, respectively, within the four-year vaccination coverage period. The proportion of HPV31 C lineage infections showed a 22.17% increase in the vaccinated group, while that of the HPV58 A2 sublineage showed a 12.96% increase. T267A and T274N in the F-G loop of HPV31 L1 protein, L150F in the D-E loop, and T375N in the H-I loop of HPV58 L1 protein were identified as high-frequency escape-related mutations. Conclusions: Differences in epidemic lineage changes and dominant mutation accumulation may result in a proportional difference in trends of HPV infection. New epidemic lineages and high-frequency escape-related mutations should be noted during the vaccine coverage period, and regional epidemic variants should be considered during the development of next-generation vaccines.

## 1. Introduction

Human papillomavirus (HPV) is a small double-stranded DNA (dsDNA) virus that causes >6 million cases of cervical and other epithelial cancers globally each year [1,2]. The genome of HPV includes a long control region (LCR), two structural protein (L1 and L2)-coding regions, and six early protein (E1, E2, E4, E5, E6, E7)-coding regions [3]. Due to the conservation of the L1 gene, the classification and the typing of papillomavirus are based on the sequence similarity of the L1 gene. Papillomaviruses with an L1 gene sequence similarity of less than 60% are classified into different genera, and all known HPVs at this stage belong to the genus *Alphapapillomavirus, Betapapillomavirus, Gammapapillomavirus, Mupapillomavirus* and *Nupapillomavirus*. Papillomaviruses with L1 gene sequence similarities between 60% and 70% are considered the same species, while within the same species, those with a similarity of less than 90% are considered different genotypes [4]. At present, there are 231 genotypes of HPVs certified by the International HPV Reference Center (www.hpvcenter.se, accessed on 5 February 2024). With the development of sequencing technology, more detailed classifications of HPVs have been identified. HPV genomes with sequence differences between 1% and 10% are classified as the same lineage, while those with sequence differences between 0.5% and 1% are classified as the same sublineage [5]. At present, 12 high-risk HPV types, closely related to carcinogenesis, have been identified as belonging to the α genera, while other low-risk types with low carcinogenic potential have been identified as low-risk HPV types [6]. Among them, HPV52 and HPV58 have a high incidence in China that is different from most parts of the world [7].

Three widely recognized HPV vaccines were launched in China in 2016, 2017, and 2018, respectively. Bivalent (Cervarix^®^, GlaxoSmithKline, London, UK) and quadrivalent (Gardasil^®^, Merck Sharp & Dohme, Branchburg, NJ, USA) vaccines targeted oncogenic HPV 16 and 18, while nonvalent (Gardasil^®^9, Merck Sharp & Dohme, Branchburg, NJ, USA) vaccines targeted five additional genotypes, including HPV 31, 33, 45, 52, and 58 [8]. The main active ingredient of HPV preventive vaccines is virus-like particles (VLPs) self-assembled from 360 L1 proteins [9], which can induce a strong antibody response against the viral capsid [10,11,12]. Antibodies can penetrate the blood vessel wall and reach a concentration that is multiple times higher in local epithelial tissue than in a natural infection state. When HPVs contact basal cells through wounds on the mucosal epithelium, antibodies located in the epithelial tissue can bind to the virus and exert a neutralizing effect. The five randomly coiled segments (B-C, D-E, E-F, F-G, H-I rings) on the L1 protein monomer are exposed to the outer surface of VLPs, becoming the main immune targets for inducing specific immune responses and neutralizing antibody binding [13]. Long-term follow-up studies have shown that the HPV vaccine has a 10-year protective effect against approximately 95% of emerging infections of the covered genotypes, and the protective effect on HPV-related squamous intraepithelial lesions is 95–100% [12,14,15]. According to WHO statistics, the application rate of the HPV vaccine is increasing, which has contributed to a decrease in the incidence of cervical cancer globally [16]. Epidemiological studies have also confirmed that vaccination significantly reduces the risk of HPV-related diseases, especially invasive cervical cancer [17,18].

However, viruses are evolutionarily autonomous and fast [19]. The increasingly widespread vaccination has undoubtedly provided new selection pressure and evolutionary dynamics for HPV. The capsid protein, as the main component of viral vaccines, becomes the main target for viruses to obtain mutations and evade immune recognition. At the same time, as the main receptor recognition site of the virus and a drug target for antiviral therapy [20], mutations in key amino acid sites on the capsid protein may enable the virus to gain long-term stable evolutionary advantages and achieve widespread accumulation in the population. Mutations in the G145 site of the hepatitis B virus surface protein (S protein) are increasingly found in breakthrough infections and often lead to fulminant, acute, and chronic hepatitis B. A study in Taiwan showed that the G145R mutant strain accounted for up to 15.4% of vaccinated children infected with the virus [21]. Further molecular biology experiments show that the widely used hepatitis B vaccine has almost no cross-protection effect on G145R mutants [22]. The capsid protein L1 of HPV is also a key immune inducer in HPV vaccines. Therefore, some HPV genotypes, lineages, sublineages, and variants with immune resistance to vaccines may be accumulating evolutionary advantages. Many studies have focused on HPV’s genetic diversity and predicted amino acid substitutions associated with vaccine escape [23,24]. A recent review evaluated 26 articles on the genetic diversity of the L1 gene, and predicted the impact on vaccine efficacy by locating non-synonymous mutation sites. These mutations include T267A and T274N in HPV31; T266K and K135R, etc., in HPV33; and N82T and V144I, etc., in HPV58 [25]. However, no definitive clinical studies have characterized genetic changes in HPV that occur during the vaccine coverage period. Recent research has also reported significant differences in the neutralizing sensitivity of serum from vaccinees to different HPV lineages [26,27]. During in vitro experiments, HPV52 D and HPV58 C lineages exhibited a >4-fold reduction in their neutralizing sensitivities of the nine-valent vaccine immune serum compared to their consensus A1 variant [28]. However, clinical evidence linking lineage displacement of HPV with vaccine escape remains to be elucidated. Therefore, this study aimed to provide clinical evidence of lineage replacement and genetic changes of HPV during the vaccine coverage period, and characterize those changes based on data from the largest cervical disease diagnosis and treatment center in China.

## 2. Methods

### 2.1. Study Population and Sample Collection

This study consisted of two stages. First, all patients visiting the Outpatient Department of the Obstetrics and Gynecology Hospital of Fudan University from March 2018 to March 2022 were included. The inclusion criteria in first stage included the following: (1) undergoing a HPV typing test; (2) positivity for any type of HPV. The number of patients finally included in the first stage was 90,583. All reports of positive HPV typing tests from these patients were collected through medical records. The overall proportions per year of HPV16, 18, 31, 33, 45, 52, and 58 were analyzed. Magnitude and continuity in the proportion of changes to the rate of infection were comprehensively considered in order to determine trends. Thus, HPV types—the proportions of which were classified according to an infection rate of ≥2%, reduction or amplification of ≥0.4%, and proportionate trending—were selected for further research. After preliminary statistics, we finally selected HPVs 31, 33, 52, and 58 for further analysis.

In the second stage, patients visiting the cervical disease clinic of the Obstetrics and Gynecology Hospital of Fudan University from November 2020 to August 2023 were further included. The inclusion criteria included (1) positivity for any type of HPV31, 33, 52, or 58; (2) complete vaccination, or nonreceipt of any HPV vaccine; (3) definite medical history and vaccination records; and (4) volunteering for this program. The number of patients finally included in the second stage was 1076. According to vaccination records, patients were divided into vaccinated and unvaccinated groups during the following analysis. A survey questionnaire and a cervical exfoliative cytology collection were required for every volunteer. The questionnaire was conducted face-to-face by the attending physician, and included (1) HPV vaccination type; (2) time to receive the third dose of vaccine (3); time of first diagnosis of HPV infection; (4) previous treatment experience; (5) ThinPrep cytologic test (TCT) results; (6) genotype of HPV infection; and (7) pathologic diagnosis of biopsy by colposcopy, which was performed by an attending physician. The cervical exfoliative cytology sample was collected by two physicians qualified for colposcopy examination and stored in phosphate-buffered saline (PBS) at −80 °C before DNA extraction. The study was approved by the hospital’s ethics committee.

### 2.2. Viral DNA Amplification and L1 Gene Sequencing

Genomic DNA was extracted from the cervical exfoliated cells using the TIANamp Genomic DNA Kit (TIANGEN). Primer sequences of HPV31, 33, 52, and 58 L1 proteins are shown in Table 1.

PCR reactions were carried out in a 50 μL reaction volume containing 1 unit of TaKaRa Ex Taq (Takara, Japan), 1 × Ex Taq Buffer (Mg^2+^ plus), 200 μM of dNTP Mixture, and 20 pmol of each primer. Thermal cycling parameters were 98 °C for 10 s, 72 °C for 90 s, and 55 °C for 10 s with 35 cycles. DNA amplicons were sequenced using the ABI 3730xl DNA Analyzer. All samples were subjected to repeated PCR and sequencing from both directions to exclude PCR artifacts.

### 2.3. Phylogenetic Analysis and Sequence Alignment

Neighbor-joining tree and evolutionary distance calculations based on the L1 gene were performed with MEGA software (version 11.0) using the Kimura 2-parameter model. The number of bootstrap replications was set at 1000. All reference sequences constructed in the phylogenetic branches were downloaded from the NCBI GenBank database and are shown in Table 2. DNA and amino acid sequence mutation analysis and alignment were also performed using MEGA software (version 11.0). Genetic distance was calculated for sequence similarity comparison. Starting from the first codon, the variance estimation method is bootstrap, repeated 1000 times; the calculation model is *p*-distance; and the base replacement information used is d: Transitions + Transformations. PhyloSuite (version 1.2.3) [29] was adopted to merge the same sequences.

### 2.4. Homology Modeling

Homology modeling was carried out using the I-TASSER (Iterative Threading ASSEmbly Refinement) online service [30] to identify loops outside the HPV VLPs in the 3-dimensional structures of L1 proteins. The L1 proteins of HPV31, 33, 52, and 58 A1 sublineages were all modeled, and amino acid mutations were mapped on the A1 L1 structures. The Swiss-PDB viewer (v4.1.0; Deep View) was used for structure analysis and mutation site localization. B-factor was combined for the comparison of mutated and non-mutated protein structures.

### 2.5. Statistical Analysis

According to the time of a patient’s first diagnosis of HPV infection, as noted on the questionnaire, we calculated the duration of infection. Then, persistent infection >24 months was defined as a categorical variable. Categorical variables were presented as a number and a percentage, and differences were assessed by Pearson’s chi-squared test and Fisher’s exact test. Quantitative variables were assessed by Student’s *t*-test. All data analyses were carried out using SPSS (version 20.0) and GraphPad Prism (version 9.5.1). A *p*-value of < 0.05 was considered a statistically significant difference.

## 3. Results

### 3.1. Trends in the Proportions of Vaccine-Covered Carcinogenic HPVs

A total of 90,583 HPV typing results from the Obstetrics and Gynecology Hospital of Fudan University were analyzed, and trends in the proportions of vaccine-covered carcinogenic HPV genotypes are shown in Figure 1. The overall proportion of HPV31 infection increased from 4.46% to 5.69%, and that of HPV58 increased from 13.20% to 13.71%. The overall proportion of HPV33 infection decreased from 5.38% to 4.96%, and that of HPV52 decreased from 22.75% to 21.32%. We excluded HPV45 owing to its low mean infection rate of 1.59%. HPV16 was also excluded from further analysis because of its fluctuating trend, and HPV18 was excluded because of its stable trend (minimum reduction of 0.31%). A continuous increasing trend of the HPV31 and 58 infections’ proportions and a continuous decreasing trend of the HPV33 and 52 infections’ proportions were identified. Finally, we collected cervical exfoliated cell samples from vaccinated or unvaccinated patients with any type of HPV31, 33, 52, or 58 infection for subsequent sequencing analysis.

### 3.2. Characteristics of HPV31-, 33-, 52-, and 58-Positive Patients

A total of 1076 patients admitted to the cervical disease clinic of the Obstetrics and Gynecology Hospital of Fudan University from November 2020 to August 2023 were involved in this study. After viral DNA amplification and gene sequencing, we finally obtained 601 L1 sequences. According to medical record information, these sequences came from 571 patients. The patients’ characteristics are shown in Table 3. Among the 601 sequences, 61 were HPV31 L1 sequences (8 from vaccinated patients); 92 were HPV33 sequences (13 from vaccinated patients); 281 were HPV52 sequences (40 from vaccinated patients); and 167 were HPV58 sequences (24 from vaccinated patients). We further counted the third dose of vaccine administration time for all vaccinated patients and calculated the vaccination duration based on the sample collection time. The mean vaccination duration was 2.09 ± 2.43 years and the median was 1.50 (2.19) years.

### 3.3. Phylogenetic Analysis of L1 Sequences

Based on patient ID number, we matched sequence information with the patient’s vaccination status and divided them into vaccinated and unvaccinated groups. Phylogenetic analysis and lineage identification were conducted separately (Figure 2). Differences in the proportions of lineages or sublineages between vaccinated and unvaccinated groups are shown in Figure 3. The proportion of infection with the HPV31 C lineage was 22.17% higher in the vaccinated group than in the unvaccinated group. Additionally, 61.54% of HPV33 L1 sequences were identified as the A1 sublineage in the vaccinated group, but 75.95% were identified in the unvaccinated group. Only one HPV33 L1 sequence belonging to the B lineage was found in the vaccinated group. More than 95% of HPV52 L1 sequences were classified as B lineage (B2 sublineage), both in vaccinated and unvaccinated groups, with very few sequences belonging to the A, C, and D lineages. All HPV58 L1 genes were identified as A lineage. The proportion of infection with the HPV52 A2 sublineage was 12.96% higher in the vaccinated group than in the unvaccinated group. All four sequences belonging to the A3 sublineage were found in the unvaccinated group. Furthermore, the HPV33 A3 sublineage and the HPV58 A1 sublineage were significantly correlated with >24-month persistent infection (*p* < 0.05).

### 3.4. Sequence Alignment and Protein Structure Analysis

We aligned all L1 sequences separated from vaccinated patients with the corresponding HPV type’s L1 standard sequence of A1 sublineage, and identified synonymous and nonsynonymous mutations (Figure 4). According to the L1 protein pentamer structure constructed based on homologous modeling, nonsynonymous mutations occurring on the outer surface of VLPs were confirmed (Figure 5), as well as the frequency of these mutations. They were as follows: I181L (E-F loop, 1/8), T267A (F-G loop, 6/8), and T274N (F-G loop, 7/8) in the HPV31 L1 sequence; T56N (B-C loop, 5/13) and G133S (D-E loop, 5/13) in the HPV33 L1 sequence; S83G (B-C loop, 1/40), N207T (E-F loop, 1/40), G308K (F-G loop, 1/40), K318T (H-I loop, 1/40), and S384D (H-I loop, 1/40) in the HPV52 L1 sequence; and L150F (D-E loop, 11/24) and T375N (H-I loop, 9/24) in the HPV58 L1 sequence.

## 4. Discussion

Several countries have included HPV vaccines in their national immunization plans since 2007. With the promotion of vaccination, cases of HPV reinfection among vaccinated populations have been reported. These data vary greatly between different countries. In 2007, Australia included the tetravalent vaccine in its immunization program. In 2021, research showed that the HPV16/18/6/11 infection rate among vaccinated individuals was 0.7%, while the non-vaccinated population during the same period was 5.5% [31]. In 2008, the UK included bivalent vaccines in its immunization program. In 2016, a study showed that the HPV16/18 infection rate among vaccinated individuals was 11.0%, while the rate for the non-vaccinated population during the same period was 28.0% [32]. At present, the only available epidemiological statistical data on breakthrough infections follows HPV vaccination, and there is a lack of in-depth mechanism research to explain this phenomenon.

To provide clinical evidence of lineage replacement and genetic changes to HR-HPV during the period of vaccine coverage and characterize those changes in eastern China, this project first analyzes the typing results of 90583 cases of HPV in the Obstetrics and Gynecology Hospital affiliated with Fudan University from 2018 to 2022, and summarizes the trend of HR-HPV proportion changes in vaccine coverage types. To further explore the reasons for this trend, this study included 1076 HPV31-, 33-, 52-, and 58-positive patients from November 2020 to August 2023, and sequenced their infected HPVs. After matching the L1 sequence information obtained from sequencing with the clinical information of the patients, the samples were divided into a vaccinated group and an unvaccinated group. Further phylogenetic analysis, sequence alignment, and protein structure analysis were also carried out.

In the four years after all three HPV vaccines were obtained in China, the proportion of infection arising from seven oncogenic HPVs covered by the vaccines showed different trends. The proportion of HPV16 infection fluctuated, while that of HPV18 remained stable, possibly because of its significantly higher carcinogenicity [33]. In addition, this phenomenon may also be attributed to the fact that this study was only carried out in China. Studies have reported that there are differences in HPV types, lineages, and sublineages prevalent in different countries and regions [34,35]. More in-depth studies have even revealed the different tendencies of amino acid substitutions in the HPV L1 gene among various regions [36]. All this evidence implies that the research on the escape and evolution of HPV requires extensive effort, on a global scale. The extremely low infection rate of HPV45 in China may have also contributed to a stable trend. However, more noteworthy trends were observed in HPV31, 33, 52, and 58. Considering the varying resistance of vaccines to different HPV lineages [26], we further explored and confirmed the differences among lineages and sublineages of HPV31, 33, 52, and 58 between vaccinated and unvaccinated patients.

The results showed a significantly decreased proportion of HPV31 A lineage and an increased proportion of C lineage in the vaccinated group, indicating the efficacy of vaccination to the A lineage, as well as the escape potential of the C lineage. Moreover, a gradual increase in the proportion of overall infection indicated that the HPV31 C lineage may have gained an evolutionary advantage and was replacing the A lineage to establish its prevalence. A similar situation was also observed in HPV33 and 58. The proportion of HPV33 A2 sublineage infection increased in the vaccinated group, suggesting that the A2 sublineage had escape potential, even if it belonged to the A lineage. The HPV33 A3 sublineage resulted in more persistent infection, which, again, may imply a survival advantage, even though its share of infection between the two groups did not proportionately increase or decrease. Considering the small differences among the A1, A2, and A3 sublineages, more discernible changes are certain to be observed when the vaccination rate is higher. The slightly increased proportion of HPV58 A2 sublineage infection in the vaccinated group also indicated its escape potential. Interestingly, while the HPV58 A1 sublineage is associated with persistent infection, it has not yet gained a survival advantage in the presence of vaccines, indicating that the evolutionary pressure exerted by vaccines on HPV may be far beyond human imagination.

The distribution pattern between HPV33 and 58 in China was also similar in that the A lineage dominated absolutely. However, the overall proportion of HPV33 infection showed a downward trend during the four years of vaccine coverage, while that of HPV58 showed an upward trend. Also, while most HPV52 L1 sequences belonged to the B2 sublineage, resulting in the absence of significant differences, the overall infection proportion of HPV52 had decreased. Collectively, these data suggest that exploration of the evolutionary pattern of HPV in the vaccination stage cannot be limited to lineages and sublineages. Therefore, we conducted further sequence alignment on all L1 sequences separated from vaccinated patients and identified several key amino acid replacement sites located on the surface of VLPs. Some of these sites have also been reported by other researchers [23,24,37,38]. As shown in Figure 4C, all amino acid substitutions that occur on the surface of HPV52 VLPs are accidental, with a frequency of 1/40, indicating that no escape-related dominant mutations have recently accumulated. This may account for the overall decreased proportion of HPV52 infection. Compared with HPV33, HPV 58 seems to have obtained specific dominant mutations with a much higher frequency of 45.83% (L150F). This may explain why the overall proportion of HPV58 infection increased, while that of HPV33 decreased. In addition, high-frequency substitutions (75% of T267A and 87.5% of T274N) were also observed in the HPV31 L1 sequence. HPV’s infection of host cells requires a complex process, in which adhesion to target cells is the first step of infection. The binding sites between the L1 protein and heparan sulfate proteoglycan (HSPG) have been confirmed to be conformational and involve residues on multiple loops (the B-C, C-D, D-E, E-F, F-G, and H-I loops). During HPV infection, most neutralizing antibodies target the primary binding site between the L1 protein and HSPG. The cryoelectron microscopy structures of HPV capsid-neutralizing antibodies (nAbs)’s complexes show that most nAbs core-binding epitopes involve the F-G loop, H-I loop, and D-E loop of the L1 protein [39]. Therefore, amino acid substitutions occurring at these positions are highly likely to affect the efficacy of neutralizing antibodies, and close attention should be paid to all these mutations to prevent their accumulation from destroying the achievements of the HPV vaccines.

Our work provides some clinical evidence linking the genetic diversity of HPV with vaccine escape variants. In the four years since the launch of all three vaccines in China, the infection rates of HPV31 and 58 have been increasing year by year, while the infection rates of HPV33 and 52 have been decreasing year by year. Further analysis revealed that the infection rates of the HPV31 C lineage, HPV33 A2 subunit, and HPV58 A2 subunit significantly increased, and multiple high-frequency amino acid replacements were located in the vaccinated group. We demonstrated that HPV may have undergone adaptive changes under the pressure of vaccination, including, but not limited to, the replacement of epidemic lineages (sublineages) and the accumulation of escape-related dominant mutations. All this evidence indicates that regional epidemic variants should be considered during the development of next-generation vaccines. However, the relatively low HPV vaccination rate in China has temporarily prevented us from conducting further research. The distribution pattern of HPV types and lineages among the Chinese population may undergo significant changes in the future. Nevertheless, the continuous observational research that we are doing is still meaningful. The changes in HPV’s genetic diversity with increasing vaccination rates will be continuously reported in the future.

## 5. Conclusions

The overall proportion of infection by HPV31 and 58 increased after vaccines became available in China, while that of HPV33 and 52 decreased visibly. Differences in the replacement of epidemic lineages and sublineages, as well as dominant mutation accumulation, may result in proportionally different trends of overall infection. The C lineage of HPV31 and high-frequency escape-related mutations of HPV31, 33, and 58 should be monitored. Regional epidemic variants should also be considered during the development of next-generation vaccines.

## Figures and Tables

**Figure 1 vaccines-12-00411-f001:**
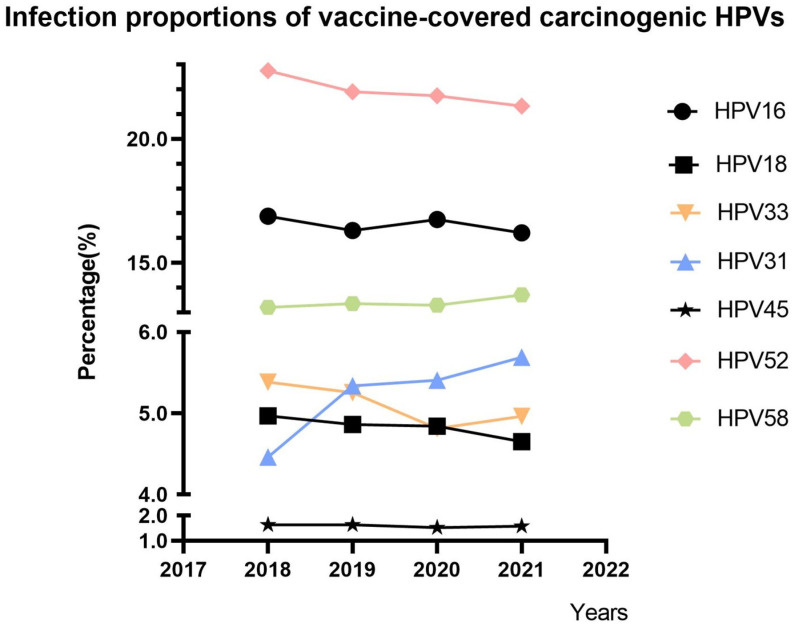
Infection trends of vaccine-covered carcinogenic HPV genotypes over time.

**Figure 2 vaccines-12-00411-f002:**
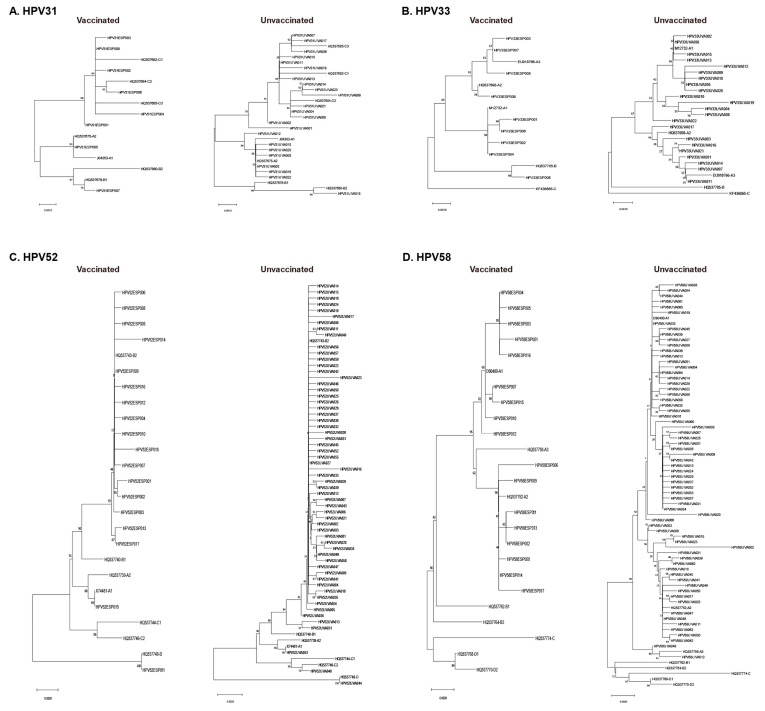
Phylogenetic tree of HPV31 (**A**), HPV33 (**B**), HPV52 (**C**), and HPV58 (**D**) L1 genes from vaccinated and unvaccinated groups.

**Figure 3 vaccines-12-00411-f003:**
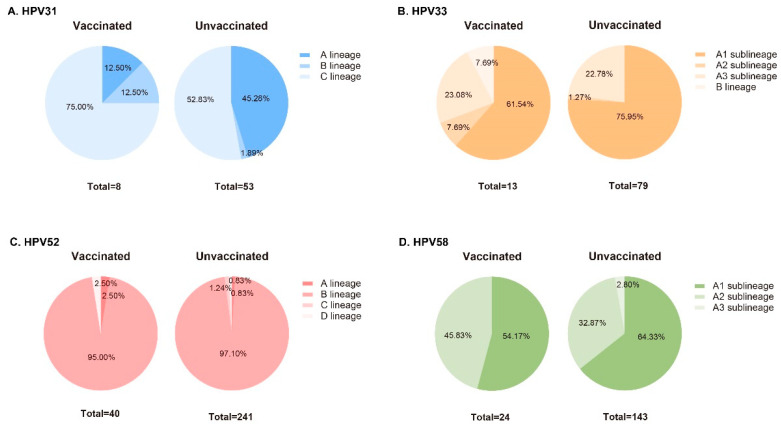
Differences in the proportions of infection by HPV31 (**A**), HPV33 (**B**), HPV52 (**C**), and HPV58 (**D**) lineages or sublineages between vaccinated and unvaccinated groups.

**Figure 4 vaccines-12-00411-f004:**
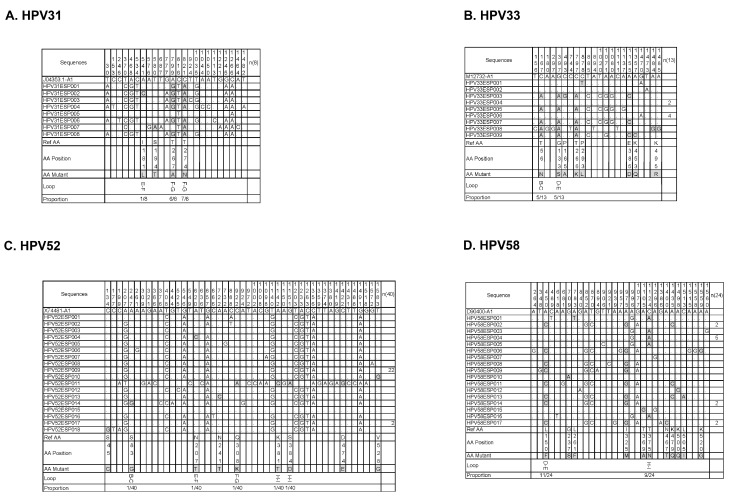
Sequence alignment of L1 gene, separated from vaccinated patients infected with HPV31 (**A**), HPV33 (**B**), HPV52 (**C**), or HPV58 (**D**), with corresponding L1 standard sequence of A1 sublineage.

**Figure 5 vaccines-12-00411-f005:**
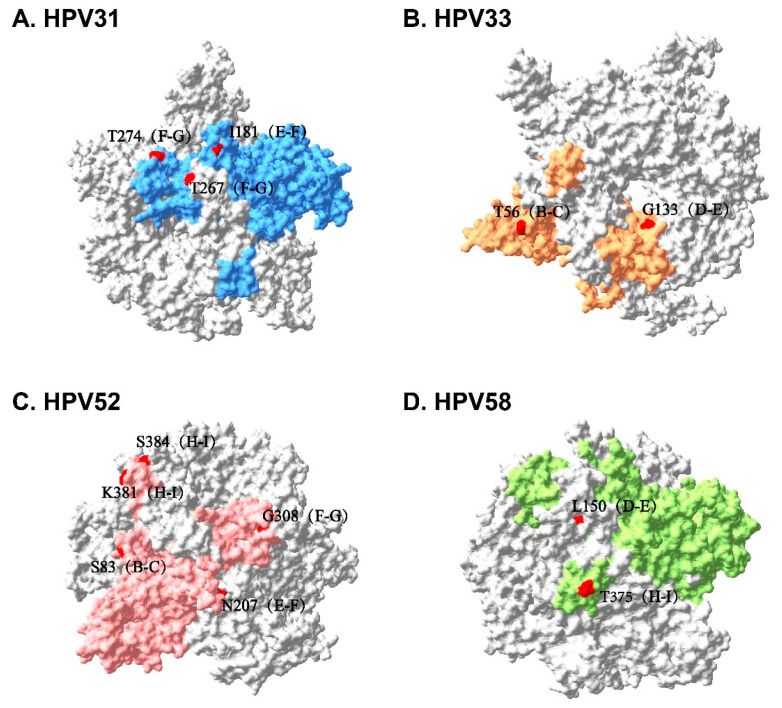
L1 protein pentamer structure of HPV31 (**A**), HPV33 (**B**), HPV52 (**C**), and HPV58 (**D**). Blue, orange, pink, and green areas are L1 monomers. Amino acid substitution sites on the surfaces of VLPs are marked in red.

**Table 1 vaccines-12-00411-t001:** E6, E7, L1, L2 gene sequence primers of HPV31, 33.

HPV Type	Gene	Primer Sequences
HPV31	L1	5′→3′ gacgtaaacgtgtatcatatttttttacag 3′ →5′ atacagacacatgtattacatacacatc
HPV33	L1	5′→3′ ggcgtaaacgttttccatatttt 3′ →5′ aacaacaacataacacaattacacaa
HPV52	L1	5′→3′ ctgacattccattaccttcgttac 3′ →5′ caatggttaccttttaacctgtgtt
HPV58	L1	5′→3′ gaacctggtccagacattgca 3′ →5′ catacaacatatacacaaacataaacaa

**Table 2 vaccines-12-00411-t002:** Reference L1 sequences of HPV31, 33, 52, and 58.

HPV Types	Lineage A	Lineage B	Lineage C	Lineage D
HPV31	A1: J04353 A2: HQ537675	B1: HQ537676 B2: HQ537680	C1: HQ537682 C2: HQ537684 C3: HQ537685	
HPV33	A1: M12732 A2: HQ537698 A3: EU918766	B: HQ537705	C: KF436865	
HPV52	A1: X74481 A2: HQ537739	B1: HQ537740 B2: HQ537743	C1: HQ537744 C2: HQ537746	D: HQ537748
HPV58	A1: D90400 A2: HQ537752 A3: HQ537758	B1: HQ537762 B2: HQ537764	C: HQ537774	D1: HQ537768 D2: HQ537770

**Table 3 vaccines-12-00411-t003:** Characteristics of HPV31-, 33-, 52-, and 58-positive patients.

Characteristics	Group	No.	%
HPV Genotype (n = 601)	HPV31	61	10.1
	HPV33	92	15.3
	HPV52	281	46.8
	HPV58	167	27.8
TCT Result (n = 558)	Normal	300	53.8
	ASC-US	125	22.4
	LSIL	92	16.5
	HSIL	23	4.1
	ASC-H	10	1.8
	SCC	1	0.2
	LSIL-HSIL	5	0.9
	HSIL-AGC	2	0.4
Vaccination (n = 565)	No	478	84.6
	Yes	87	15.4
Vaccination Type (n = 87)	Bivalent	9	10.3
	Quadrivalent	36	41.4
	Nonvalent	40	46.0
	Undefined	2	2.3
Multiple Infection (n = 501)	No	272	54.3
	Yes	229	45.7
Persistent Infection > 24 months (n = 515)	No	309	60.0
	Yes	206	40.0
Pathological Diagnosis of Biopsy (n = 544)	Normal	13	2.4
	Inflammation	203	37.3
	LSIL	261	48.0
	HSIL	64	11.8
	SCC	1	0.2
	AIS	1	0.2
	AC	1	0.2

ASCUS: atypical squamous cells of undetermined significance; LSIL: low-grade squamous intraepithelial lesions; HSIL: high-grade squamous intraepithelial lesions; ASC-H: atypical squamous cells that cannot exclude HSIL; SCC: squamous cell carcinoma; AGC: atypical glandular cells; AIS: adenocarcinoma in situ; AC: adenocarcinoma.

## Data Availability

Data available on request from the authors. The data that support the findings of this study are available from the corresponding author upon reasonable request.

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
