# Peer review of "Lineage Replacement and Genetic Changes of Four HR-HPV Types during the Period of Vaccine Coverage: A Six-Year Retrospective Study in Eastern China"

_vaccines, 2024, doi:10.3390/vaccines12040411_

Round 1
Reviewer 1 Report
Comments and Suggestions for Authors
In the manuscript entitled “Lineage replacement and genetic changes of four HR-HPV types during the period of vaccine coverage: a six-year retrospective study in eastern China” the authors present significant data concerning the lineage replacement and genetic alteration of HR-HPV genotypes HPV 31, 33, 52, 58 for the period of vaccine coverage in eastern China. The manuscript is well written and data are sufficiently presented. However, some points need to be addressed.
In introduction the authors mentioned that “Among all those genotypes, 12 are currently classified as highly carcinogenicity, including HPV 16, 18, 31, 33, 56 35, 39, 45, 51, 52, 56, 58, and 59”. The appropriate literature is missing. Moreover, it is known that the HPVs of genus alpha are classified into high-risk and low risk HPV genotypes, according to their carcinogenic potential. Please mention the low-risk HPV genotypes with the appropriate literature, as well (Virol. J. 2010;7:11.doi: 10.1186/1743-422X-7-11). In addition, a short description of current epidemiologic evidences of HPV distribution in China would help the readers to better understand the significance of the present analysis.
In materials and methods the authors are required to mention the total number of patients that enrolled in stage I and stage II, respectively. It is not clear how many vaccinated and unvaccinated individuals of stage I were used in order to estimate the proportion of HPV genotypes.
In materials and methods the authors used the Neighbor Joining methodology in order to construct the phylogenetic trees. In DNA sequences the Maximum Likelihood methodology is used with the best fit evolutionary model. The phylogenetic trees must be re-evaluated.
In results it was mentioned that “A total of 90583 HPV typing results from the Obstetrics and Gynecology Hospital of Fudan University were analyzed” Are these patients from the Outpatient department? Please clarify. Moreover, the authors are required to explain why they obtained sequences from only 571 out of 1076 patients of stage II.
The authors suggested that HPV may have undergone adaptive changes under the pressure of vaccination, while specific amino acid substitutions are detected in the examined population with high frequency, including HPV31 L1 T267A (F-G ring, 6/8), T274N (F-G ring, 7/8), L150F (D-E loop, 11/24) and T375N (H-I loop, 9/24) in the HPV58 L1 sequence. According to my opinion it would be interesting to examine whether these positions are under positive selective pressure.
· In the Discussion the authors should further describe the reasons why the proportion of HPV16 infection fluctuated. A previous study concerning a global amino acid mutation analysis of HPV16 L1 protein revealed a number of amino acid substitutions located in the L1 hypervariable surface loops and it was further discussed whether these mutations enable an escape from L1 vaccines (Viruses 2022 Dec 31;15(1):141. doi: 10.3390/v15010141). Moreover, considerable data concern the Chinese population, as well. The authors are required to further discuss their findings considering this analysis.
The Discussion must be improved as several sections are repeated. Moreover, the part of the last paragraph which describes the study design should be moved in the beginning of the Discussion in order to avoid confusion.
Reviewer 2 Report
Comments and Suggestions for Authors
The manuscript by Wenjie Qu et al. is an interesting retrospective study conducted in eastern China, aiming to analyze lineage replacement and genetic changes of four HR-HPVs during the period of vaccine coverage. Here are my suggestions for publication:
Introduction:
- I find the paragraph on the vaccine related- immune evasion of other viruses very interesting but too long for fitting in. Please shorten it
- Please add a comment on the reported literature regarding the known HPV genetic diversity and predicted amino acid substitutions associated with vaccine escape
- While the authors stated that there are no clinical evidences for lineage replacement and genetic changes of HPV during the vaccine coverage period, please comment the status of this genetic analysis at preclinical or cellular level
Methods:
- Please better describe in section 2.1 the two stages of the study and thus the two groups of patients: what’s the correlation between them, age of the patients, protocols/timings of the surveys and sample collections
Results:
- Section 3.3: what’s the statistical significance between the infections of the different lineages and between the vaccinated and non-vaccinated?
- Table 3: is it possible to analyse the influence of each type of vaccine in the HPVs- lineage replacement and genetic changes?
Comments on the Quality of English LanguageThe manuscript would benefit from a second revision to correct the construction of some sentences
Reviewer 3 Report
Comments and Suggestions for Authors
Dear Authors and Editors!
Thank you for the opportunity to review your manuscript.
Vaccination against HPV infection is a useful tool to prevent cervical and epithelial malignanacy. The dynamics of distribution of HPV subtypes is strictly required. The manuscript is actual. The Methods are are new and correspond to the study's aims. The results are clear and discussion contains the comparison of the study results with the contemporary literature. The discussion reflects the study results.
I have several minor suggestions
1) Abstract: Authors may add inforvation about the vaccines.
2) Did all patients receive vaccines, please add information about distribution of the vaccines between study population
3) What was a mean time since the last vaccination to the study inclusion?
4) Can you provide the similar to Fig 1 data about non-vaccinated patients or you can add the information to the fig.1
5) Did you see difference between HPV viruses and type of the vaccine? Can you provide the statistical analysis?
6) Did you have pateints who received vaccination being infected with HPV?
Round 2
Reviewer 1 Report
Comments and Suggestions for Authors
The authors have addressed all of my concerns. The revised version of the manuscript is suitable for publication.